# Hepatic Sam68 Regulates Systemic Glucose Homeostasis and Insulin Sensitivity

**DOI:** 10.3390/ijms231911469

**Published:** 2022-09-29

**Authors:** Aijun Qiao, Wenxia Ma, Ying Jiang, Chaoshan Han, Baolong Yan, Junlan Zhou, Gangjian Qin

**Affiliations:** 1Department of Biomedical Engineering, School of Medicine and School of Engineering, University of Alabama at Birmingham, Birmingham, AL 35294, USA; 2Zhongshan Institute for Drug Discovery, Shanghai Institute of Materia Medica, Chinese Academy of Sciences, Zhongshan 528400, China; 3Shanghai Institute of Materia Medica, Chinese Academy of Sciences, 555 Zu Chong Zhi Road, Shanghai 201203, China; 4Feinberg Cardiovascular Research Institute, Feinberg School of Medicine, Northwestern University, Chicago, IL 60611, USA

**Keywords:** Sam68, hepatocytes, gluconeogenesis, high-fat diet, diabetes, CRTC2, insulin sensitivity, glucagon, glucose metabolism, protein–protein interaction

## Abstract

Hepatic glucose production (HGP) is an important component of glucose homeostasis, and deregulated HGP, particularly through gluconeogenesis, contributes to hyperglycemia and pathology of type-2 diabetes (T2D). It has been shown that the gluconeogenic gene expression is governed primarily by the transcription factor cAMP-response element (CRE)-binding protein (CREB) and its coactivator, CREB-regulated transcriptional coactivator 2 (CRTC2). Recently, we have discovered that Sam68, an adaptor protein and Src kinase substrate, potently promotes hepatic gluconeogenesis by promoting CRTC2 stability; however, the detailed mechanisms remain unclear. Here we show that in response to glucagon, Sam68 increases CREB/CRTC2 transactivity by interacting with CRTC2 in the CREB/CRTC2 complex and occupying the CRE motif of promoters, leading to gluconeogenic gene expression and glucose production. In hepatocytes, glucagon promotes Sam68 nuclear import, whereas insulin elicits its nuclear export. Furthermore, ablation of Sam68 in hepatocytes protects mice from high-fat diet (HFD)-induced hyperglycemia and significantly increased hepatic and peripheral insulin sensitivities. Thus, hepatic Sam68 potentiates CREB/CRTC2-mediated glucose production, contributes to the pathogenesis of insulin resistance, and may serve as a therapeutic target for T2D.

## 1. Introduction

Type-2 diabetes (T2D) is characterized by the dysregulation of glucose homeostasis, resulting in hyperglycemia. It is found in 9% of the adult population worldwide and directly causes at least 1.5 million deaths annually [1]. In addition, T2D significantly increases comorbidities of other chronic health problems, including cardiovascular disease, stroke, and kidney disease, and people who live with long T2D are at increased risk of liver-related complications, including nonalcoholic fatty liver disease, severe liver scarring, liver cancer, and liver failure [2].

In healthy individuals, blood glucose levels are maintained within a narrow range through physiological balance between glucose production, about 90% by liver, and glucose consumption by peripheral organs. The two processes are tightly regulated by opposing actions of glucagon and insulin hormones. During fasting, glucagon signaling is raised to trigger hepatic glucose production (HGP) through gluconeogenesis and glycogenolysis, while basal level of insulin signaling limits glucose disposal [3]. During feeding, upregulated insulin signaling suppresses hepatic gluconeogenesis, promotes glucose disposal in the periphery, and simultaneously inhibits glucagon secretion from pancreatic α-cells, further shutting down HGP [4]. It is widely accepted that hepatic glucose overproduction, especially through gluconeogenic pathway, and reduction in glucose disposal are major causes of hyperglycemia in T2D [5].

Hepatic gluconeogenic gene expression is primarily mediated by the transcription factor cAMP-response element (CRE)-binding protein (CREB) and its coactivator, the CREB-regulated transcriptional coactivator 2 (CRTC2). CRTC2 facilitates CREB binding to the CRE motif in the promoters of gluconeogenesis genes, including PPARγ coactivator 1α (PGC-1α), a major player of gluconeogenesis, and phosphoenolpyruvate carboxykinase (PEPCK) and glucose-6-phosphatase (G6Pase), the rate-limiting gluconeogenic enzymes [6]. The CREB/CRTC2 axis is balanced by the opposing effects of insulin and glucagon. During fasting, glucagon signaling promotes gluconeogenesis via dephosphorylation of the CRTC2 [7], triggering its translocation into the nucleus, association with CREB, and thus transcription of gluconeogenic gene [8]; during refeeding, insulin signaling leads to CRTC2 nuclear export and degradation, repressing hepatic gluconeogenesis [9]. In parallel, the CREB/CRTC2 axis modulates insulin sensitivity in the liver and peripheral organs [10,11,12], and dysregulation of the CREB/CRTC2 axis is a major contributor to the hyperglycemia in T2D [1,13].

Src-associated-in-mitosis-of-68kDa (Sam68) is a member of the signal-transducer-and-activator-of-RNA (STAR) family of RNA-binding proteins [14]. Studies in last decade reveal that Sam68 is involved in multiple cellular functions, including RNA processing [15,16], growth factor receptor and kinase signaling [17,18], transcription [19,20], cell cycle regulation and apoptosis [21,22]. Interestingly, global Sam68 knockout (Sam68^–/–^) mice exhibit several interesting phenotypes, including defects in spermatogenesis [23,24], decline of body weight coupled with enhanced adipose thermogenic program [25,26,27], and reduced blood glucose levels in both fed and fasted states [27,28]. By generating hepatocyte-specific Sam68 knockout mice, our group recently discovered that Sam68 promotes HGP via enhancing CRTC2 protein stability [28]; nevertheless, whether Sam68 regulates CREB/CRTC2 transcriptional activity and systemic insulin sensitivity remains elusive [28].

In this study, we found that Sam68 promotes hepatic gluconeogenic gene expression and glucose production by interacting with CRTC2 in the CRTC2/CREB complex occupying the CRE motifs, and that glucagon promotes Sam68 nuclear import, whereas insulin elicits its nuclear export, in hepatocytes. Furthermore, Sam68 expression is significantly elevated in the livers of nutrition-restricted mice and diabetic mice, and ablation of Sam68 in hepatocytes (Sam68^LKO^) lowers blood glucose levels in HFD-induced diabetic mice and enhances systemic insulin sensitivity.

## 2. Results

### 2.1. Sam68 Expression Is Elevated in the Livers of Nutrition-Restricted Mice and Diabetic Mice

We initiated our investigation by measuring Sam68 expression in the livers of healthy mice under feeding and fasting conditions; both mRNA and protein levels of Sam68 were significantly higher under fasting condition (Figure 1A,B). Then, we assessed Sam68 expression in the livers of HFD-induced prediabetic and obese mice and normal chow diet (ND)-fed mice (Appendix A) as well as in db/db mice and db/m controls (Appendix A). Sam68 protein was markedly upregulated in HFD-induced and db/db diabetic mice than in respective control animals (Figure 1C,D). Interestingly, the Sam68 upregulation is associated with an elevated level of Sam68 protein in the nucleus, as shown in the HFD-fed mice compared to ND-fed control animals (Appendix A). Thus, hepatic Sam68 expression is associated with physiological and pathological states with increased glucose production.

### 2.2. Sam68 Promotes Hepatic Gluconeogenesis by Potentiating Glucagon Signaling

To understand the role of hepatic Sam68 expression in glucose production, we generated hepatocyte-specific Sam68 knockout mice (Sam68^LKO^) [28], in which the Sam68 gene was successfully deleted from the liver (Figure 2A). Compared to Sam68^f/f^ littermates, Sam68^LKO^ mice displayed a significantly lowered blood glucose level under both feeding and fasting conditions (Figure 2B), which was accompanied by a significant reduction in the mRNA and protein expression of key gluconeogenic genes including PGC-1α, PEPCK and G6Pase in the liver (Figure 2C,D). Nevertheless, no obvious histomorphological differences were observed between the two genotypes of animals (Appendix A). To determine if Sam68 regulates gluconeogenesis in human hepatocytes, we generated HepG2 cells that stably express Sam68 shRNA (Sam68-KD) or non-target shRNA (NT), respectively, by using lentiviral vectors. Sam68 protein expression was downregulated by 87% in Sam68-KD cells (Figure 2E), which led to a drastic attenuation in glucagon-induced glucose production (Figure 2F) and time-dependent gluconeogenic gene expression (Figure 2G). Thus, Sam68 promotes hepatic gluconeogenesis and glucose production by potentiating glucagon signaling.

### 2.3. Sam68 Potentiates Glucagon Signaling by Augmenting CREB/CRTC2 Transactivity

We have recently observed that Sam68 stabilizes CRTC2 in the basal condition in hepatocytes [28]. Here, our co-immunoprecipitation (co-IP) analyses further revealed that endogenous Sam68 and CRTC2 proteins interact in the nucleus of mouse primary hepatocytes under glucagon treatment (Figure 3A). As CRTC2 interacts with CREB and facilitate CREB binding to the CRE motif of promoters [29], we tested whether the Sam68-CRTC2 interaction promotes CREB/CRTC2 transactivity. Promoter–reporter assays were performed in HepG2 cells transfected with plasmid containing CRE–Promoter that drives luciferase expression; Sam68 overexpression dramatically increased CRE–Promoter activity (Figure 3B). Furthermore, results from chromatin immunoprecipitation (ChIP) assays indicate that Sam68 physically occupies the CRE-containing sequences of the PGC-1α, G6Pase, and PEPCK promoters in the hepatocytes under glucagon treatment (Figure 3C). Thus, Sam68 potentiates glucagon signaling through enhancing CREB/CRTC2-mediated transcription. The role of CRTC2 in Sam68-mediated gluconeogenesis was further verified by knocking down CRTC2 (CRTC2-KD) in Sam68-overexpressing HepG2 cells (Figure 4A,C); while Sam68 overexpression markedly increased glucagon, forskolin or Bt2-cAMP-induced glucose production and gluconeogenic gene expression, these effects were completely diminished by CRTC2-KD (Figure 4D,E), confirming that Sam68 promotes hepatic gluconeogenesis via CRTC2.

### 2.4. Glucagon Promotes Sam68 Nuclear Import, Whereas Insulin Elicits Its Nuclear Export

Consistent with the observation that insulin induces Sam68 cytoplasmic translocation in rat adipocytes [30,31], we found in mouse primary hepatocytes that, after insulin treatment, Sam68 abundance was increased in the cytoplasm but decreased in the nucleus, while total Sam68 protein amount was unchanged (Figure 5A). Interestingly, contrasting effects (i.e., Sam68 abundance was increased in the nucleus but decreased in the cytoplasm) were observed when the cells were treated with glucagon (Figure 5B). Furthermore, we analyzed Sam68 protein levels in the cytoplasmic and nuclear fractions of liver tissues from mice that had received intraperitoneal injection of insulin or glucagon. Our results confirmed that insulin induces Sam68 translocation from the nucleus to cytoplasm (Figure 5C) and glucagon elicits Sam68 translocation in the opposite direction (Figure 5D) in vivo in the physiological state.

### 2.5. Ablation of Hepatic Sam68 in Mice Leads to Improved Systemic Insulin Sensitivity

The finding of upregulated Sam68 expression in the liver of diabetic mouse models promoted us to examine whether deletion of hepatocyte Sam68 attenuates hyperglycemia. Experiments were conducted in Sam68^LKO^ mice and Sam68^f/f^ littermates that had been fed HFD for 12 weeks. Compared to Sam68^f/f^ controls, Sam68^LKO^ mice exhibited significantly lowered blood glucose levels under both feeding and fasting conditions (Figure 6A) and improved insulin tolerance (Figure 6B). Furthermore, we assessed the insulin signaling by measuring insulin induced AKT phosphorylation at serine 473 and tyrosine 308, which was significantly increased in Sam68^LKO^ mice, not only in the liver (Figure 6C) but also in peripheral tissues including epiWAT (Figure 6D) and skeletal muscles (Figure 6E), indicating improved systemic insulin sensitivity.

## 3. Discussion

Uncontrolled glucagon signaling and blunted insulin sensitivity contribute to the increased HGP, particularly gluconeogenesis, by the liver and the decreased glucose disposal by peripheral tissues, which are primary causes of T2D [1,32,33]. Extended from our recent observation that Sam68 potently promotes HGP by increasing CRTC2 stability [28], this current study further revealed that hepatic Sam68 maintains systemic glucose homeostasis by interacting with CRTC2 to promote CREB/CRTC2 transactivity and gluconeogenic gene expression, and that glucagon promotes Sam68 nuclear import, whereas insulin induces its nuclear export. In diabetes mouse models, Sam68 levels in the liver are drastically upregulated, contributing to the hyperglycemia; the ablation of hepatic Sam68 lowers blood glucose level and improves insulin sensitivity (Figure 7). Thus, targeting Sam68 action may be a novel strategy for T2D.

The central role of the hepatic CREB/CRTC2 axis on gluconeogenesis has been well recognized [9,12,13,34,35,36]. The CREB/CRTC2 axis has been intensively investigated for T2D drug development [9,12,37,38]. It is known that CRTC2 activity is inversely correlated with systemic insulin sensitivity [9,12,34], whether the improved insulin sensitivity in Sam68^LKO^ mice is due to attenuated CRTC2 activity remains a subject of our further investigation.

We found that Sam68 protein shuttles between the cytoplasm and nucleus in hepatocytes, which appears to be modulated by the opposing actions of insulin and glucagon. Notably, a number of posttranscriptional modifications, including tyrosine phosphorylation [39], serine/threonine phosphorylation [40], acetylation [41], methylation [42] and SUMOylation [43], have been reported to be associated with Sam68 subcellular localization. Thus, it is likely that diverse nutritional states and metabolic regulators alter the posttranslational modifications of Sam68 to influence glucose metabolism. Studies to identify these site-specific modifications and the associated functions may provide additional mechanistic insights into T2D pathogenesis. In addition, Sam68 has been shown to regulate numerous cellular functions through integrating multiple signaling pathways (e.g., growth factors, inflammation) [44,45,46]; thus, it may act to overlay environmental factors to modulate glucose metabolism, insulin sensitivity and the pathogenesis of T2D.

In summary, hepatic Sam68 interacts and stabilizes CRTC2, augments CREB/CRTC2-mediated gluconeogenic gene expression and HGP; deletion of hepatic Sam68 lowers blood glucose levels and improves insulin sensitivity in diabetic mouse models. Thus, Sam68 may be targeted to improve diabetic hyperglycemia in humans.

## 4. Materials and Methods

### 4.1. Animal Studies

All animal experiments in this report were approved by the Institutional Animal Care and Use Committee (IACUC) of the University of Alabama at Birmingham and comply with all relevant ethical regulations (IACUC-21712), including the National Institutes of Health (NIH) “Guide for the Care and Use of Laboratory Animals”. Experiments were conducted in 8–12-week-old, male, C57BL/6 mice unless otherwise specified. Animal numbers for the experiments were pre-calculated based on the reported similar studies from us and other laboratories with 80% power [28,32,47]. The basic characteristics of our model animals were reported, and all physiological and biochemical measurements were performed in double-blinded fashion. Mice were fed ad libitum and maintained under a 12:12-h light: dark cycle.

### 4.2. Genetic Mice and Diabetic Mouse Models

Sam68^−/−^ mice were obtained from Dr. Stéphane Richard [48]. Sam68^f/f^ mice were generated as we have recently reported and donated to The Jackson Laboratory Repository (JAX Stock No. 037100) [28]. Sam68^f/f^ mice were crossed with mice carrying Albumin-cre (The Jackson Laboratory, No. 003574) to generate Alb-Cre^+^; Sam68^f/f^ (hepatocyte-specific Sam68 knockout or Sam68^LKO^) mice and Sam68^f/f^ littermate controls. The sequences of primers used for PCR genotyping are reported in Appendix A. To induce obese and prediabetic mouse model, four-week-old mice were fed with HFD (58% of energy from fat; Research Diet, New Brunswick, NJ, D12331,) for 12 weeks. The db/db and db/+ mice were purchased from Jackson Laboratory (No. 000642). Whereas C57BL/6J mouse strain is frequently used to model diet-induced obesity and insulin resistance (prediabetes) for their reliable weight gain on high-fat diet (HFD), db/db (leptin deficiency) mouse is the most widely used animal model of type 2 diabetes mellitus (T2DM) and sustains hyperglycemia phenotype after 8 weeks old.

### 4.3. Blood Glucose Measurements

Blood glucose levels were measured from tail bleeds using an automated glucose monitor (Bayer Contour, Whippany, NJ, USA). For overnight fasted treatment, the mice were denied food but with free access to water. For insulin tolerance test (ITT), mice were fasted for 6 h, followed by intraperitoneal injection of insulin (1 U/kg body weight). Then, blood glucose levels were measured from tail bleeds 0, 15, 30, 45, 60, 90, and 120 min after injection, as we previously described [49].

### 4.4. Lentiviral Vectors

Lentiviral plasmids Sam68-shRNA (SHCLNV-NM_006559) and non-targeting-shRNA (SHC016) were purchased from Sigma (Saint Louis, MO, USA). Lentiviral vectors were produced using the LentiSuite Basic Kit (System Biosciences, Palo Alto, CA, USA, LV340A-1) following manufacturer’s protocol.

### 4.5. HepG2 Cells Culture and Lentivirus Infection

HepG2 cells (CRL-10741) were obtained from ATCC (Manassas, VA, USA) and cultured following manufacturers’ instructions. The cells were infected with Sam68-shRNA or non-targeting-shRNA lentivirus vector; 48 h later, the transduced cells were selected in puromycin (7.5 ug/mL) for 14 days prior to experiments.

### 4.6. Mouse Primary Hepatocyte Isolation

Mouse primary hepatocytes were isolated via a two-step perfusion procedure using liver perfusion media (Krebs–Ringer Biocarbonate Buffer, Sigma, K4002) and digestion buffer (Krebs–Ringer Biocarbonate Buffer with 0.1–0.15% collagenase) as we previously described [50]. After isolation, cells were cultured on collagen-coated plates in completed DMEM containing 10% FBS, and 1% penicillin and streptomycin. After 6 h of attachment, the medium was replaced, and the cells were incubated overnight prior to subsequent experiments.

### 4.7. Plasmids, Cell Transfection and cAMP Response Element (CRE) Luciferase Activity

Plasmids pcDNA3-HA (128034), pcDNA3-HA-Sam68 (17690), pcDNA3-Flag-CRTC2 (22975), and pCF-Flag-CREB (22968) were purchased from Addgene (Watertown, MA, USA). The CRE luciferase reporter system, including the reporter plasmid with a promoter that contains CRE and drives firefly luciferase expression (CRE–Promoter–Luciferase) and a control plasmid with CMV promoter that drives renilla expression, was purchased from Qiagen (CRE Luciferase Assay Kit, Hilden, Germany, CCS-002L). For assessment of Sam68 and CRTC2-mediated CRE activation, the CRE luciferase reporter plasmids were co-transfected in HepG2 cells along with pcDNA3-HA-Sam68 or pcDNA3-HA, and/or Flag-CRTC2 and/or pCF-Flag-CREB plasmids by using Lipofectamine 3000 Transfection Reagent (Invitrogen, Inc., Waltham, MA, USA). Forty-eight hours later, luciferase activity was evaluated with the Dual-Luciferase Reporter Assay System (Promega, Madison, WI, USA, E1910) and normalized to renilla activity following manufacturer’s protocol as we described previously [20].

### 4.8. Cytoplasmic and Nuclear Fractionation

Cytoplasmic and nuclear fractions of mouse hepatocytes or liver tissue were isolated using NE-PER™ Nuclear and Cytoplasmic Extraction Reagents (Thermo Scientific™, Waltham, MA, USA, 78835). For assessments of glucagon- or insulin-induced Sam68 subcellular localization in vivo, mice were fasted for 4 h, followed by intraperitoneal injection of glucagon (30 μg/kg body weight, Sigma, G1774) or insulin (1 U/kg body weight, Sigma, I19278); the liver tissues were isolated 10 min after glucagon injection or 20 min after insulin injection, and the cytoplasmic and nuclear fractions were isolated and the Sam68 protein abundance was evaluated by Western blotting.

### 4.9. Western Blotting

Western blotting was performed in cytoplasmic and nuclear fractions or in cellular protein extracts. For protein extraction, 1 × 10^7^ cells or 100 mg tissue were homogenized in 1 mL RIPA lysis buffer (50 mM Tris-HCl pH 8.0, 1 mM EDTA, 1% Triton X-100, 0.1% SDS, 150 mM NaCl) that contains protease-inhibitor (Sigma, 4693132001) and phosphatase-inhibitor (Sigma, 4906837001) cocktail. After incubation with agitation for 30 min at 4 °C, the samples were centrifuged at 13,000 rpm, 4 °C, for 10 min. Then, the supernatant was collected, and the protein concentration was determined by bicinchoninic acid (BCA) assay (Thermo Fisher Scientific, Waltham, MA, USA; 23228). For immunoblotting, proteins were denatured by heating at 95 °C for 10 min, separated by SDS-PAGE, then transferred onto a PVDF membrane (Bio-Rad, Hercules, CA, USA, 1620174). After blocking with 5% non-fat milk in tris-buffered saline (TBS) for 1 h, the membrane was incubated with primary antibody in 3% BSA-containing TBS overnight at 4 °C, washed 4 times with TBST (TBS containing 0.5‰ Tween 20), incubated with secondary antibody at room temperature for 1 h, washed 4 times with TBST, and then developed with Amersham™ ECL™ Prime Western Blotting Detection Reagent (Healthcare, Chicago, IL, USA, RPN2232GE). Protein signals were imaged with a Bio-Rad ChemiDoc System (Bio-Rad, Hercules, CA). Antibodies are listed in Appendix A.

### 4.10. Co-Immunoprecipitations (Co-IP)

Immunoprecipitation (IP) was performed as previously described [28]. Briefly, hepatocytes were lysed in 1% NP40 lysis buffer (50 mM Tris-HCl pH 8.0, 1% Triton X-100, 150 mM NaCl, 0.25% sodium deoxy cholate and protease inhibitor cocktail). Samples were constantly agitated for 30 min at 4 °C, then centrifuged at 13,000 rpm for 10 min at 4 °C, and the protein concentration in the supernatant was determined. Then, 1 mg total protein was incubated with protein A/G plus agarose conjugated Sam68 antibody (Santa Cruz, CA, USA) at 4 °C for overnight and washed 4 times with lysis buffer; the immunoprecipitants were eluted by boiling for 5 min, and extracts were analyzed by Western Blotting.

### 4.11. Chromatin Immunoprecipitation (ChIP)

ChIP experiments were performed using a ChIP assay kit (EMD Millipore, Burlington, MA, USA,17-295) following manufacturer’s instructions. Briefly, cells were fixed in 1% formaldehyde for 10 min at room temperature, then quenched by adding glycine to a final concentration of 125 mM. Cells were subsequently washed twice with PBS containing a protease inhibitor cocktail (1 mM PMSF,1 μg/mL aprotinin, and 1 μg/mL pepstatin A), pelleted, resuspended in 200 μL ChIP lysis buffer (1% SDS, 10 mM EDTA, and 50 mM Tris, pH 8.1), and sonicated in an ultrasonic processor (Sonicator 3000; Misonix, Farmingdale, NY, USA) to shear the DNA into ~500 bp fragments. After precleared with protein A agarose beads, samples were immunoprecipitated with Sam68 antibody or mouse IgG at 4 °C overnight. The immunoprecipitated DNA–protein complexes were eluted from the agarose beads twice by adding elution buffer (1% SDS, 0.1 M NaHCO3, pH 8.0) at room temperature for 15 min, and the cross-linking was reversed by heating at 65 °C for 2 h. The immunoprecipitated DNA was analyzed for Sam68 occupancy on the CRE in the promoters of PGC-1α, PEPCK, and G6Pase by qRT-PCR. The primers used are listed in Appendix A.

### 4.12. Quantitative Real-Time Polymerase Chain Reaction (qRT-PCR)

Total RNA was isolated with TRIzol Reagent and reverse transcribed into cDNA with Reverse Transcription Reagents (Applied Biosystems, Waltham, MA, USA, 4368814); then, mRNA levels of gene were determined by qPCR (ABI3000; Applied Biosystems) with SYBR Green Real-Time PCR Master Mix (Applied Biosystems, 4309155). Duplicate reactions were performed for each sample, and the relative mRNA expression for each gene was calculated via the 2(−ΔΔCt) method and normalized to β-actin gene expression. The primers are listed in Appendix A.

### 4.13. Glucose Production

HepG2 cells were seeded on six-well plates (1 × 10^6^ cells/well), cultured overnight in DMEM with 10% FBS, then washed three times with PBS and incubated in glucose-free DMEM pH 7.4 without phenol red and supplemented with 20 mM sodium lactate, 2 mM sodium pyruvate, and 1 mM glycerol in the presence of 100 nM glucagon or PBS; 4 h later, 0.5 mL medium was collected, and the glucose concentration was measured with a colorimetric glucose assay kit (Eton Bioscience, Inc., San Diego, CA, USA, 120007200P). Readings were normalized to the total protein content in whole-cell lysates, as previously described [49].

### 4.14. Analysis of In Vivo Insulin Signaling

The mice on HFD for 3 months were fasted for 16 h, then intraperitoneally injected with insulin (1 U/kg) or saline; 20 min later, mice were euthanized, and the liver, epiWAT and skeletal muscle tissues were quickly excised, snap frozen in liquid nitrogen, and stored at –80 °C until use. For evaluating insulin signaling, tissues were homogenized in RIPA buffer containing protease and phosphatase inhibitors (Sigma), then tissue proteins were extracted and analyzed via Western blotting with primary antibodies against Ser-473-AKT, Thr-308-AKT and total AKT (Appendix A) as we previously described [25].

### 4.15. Liver Histology

Hematoxylin and eosin (H and E) staining was performed in the liver tissue collected from Sam68^f/f^ and Sam68^LKO^ mice and fixed in 4% paraformaldehyde (PFA) by following the protocol reported by Dai et al. [51]. For assessing Sam68 expression, liver tissues were collected from HFD- or ND-fed mice, and cryosections were subjected to immunofluorescence staining. Briefly, the sections were blocked in 10% donkey serum in PBST at room temperature for 1 h, then incubated with Sam68 antibody (1:100) for overnight at 4 °C. Next day, the sections were washed with PBST, followed by incubation with Alexa Fluor 488-conjugated Donkey anti-rabbit IgG secondary antibody (Invitrogen, Waltham, MA, USA) (Green, 1:300) for 1 h at room temperature in dark. Nuclei were counterstained with DAPI (blue, Vectashield, Antifade Mounting Medium with DAPI). The fluorescence signal was examined under Nikon NI-S-E Microscope (Nikon America, Inc., Melville, NY, USA).

### 4.16. Statistical Analysis

Data are presented as mean ± standard error of the mean (s.e.m.). The normality of data was tested using the Shapiro–Wilk test. Only the data with normal distributions were applied to parametric tests; statistical significance between two groups was evaluated via the unpaired two-tailed Student’s *t* test and among 3 or more groups via one-way or two-way analysis of variance (ANOVA) with one or two independent variables. A *p*-value of less than 0.05 was considered significant.

## Figures and Tables

**Figure 1 ijms-23-11469-f001:**
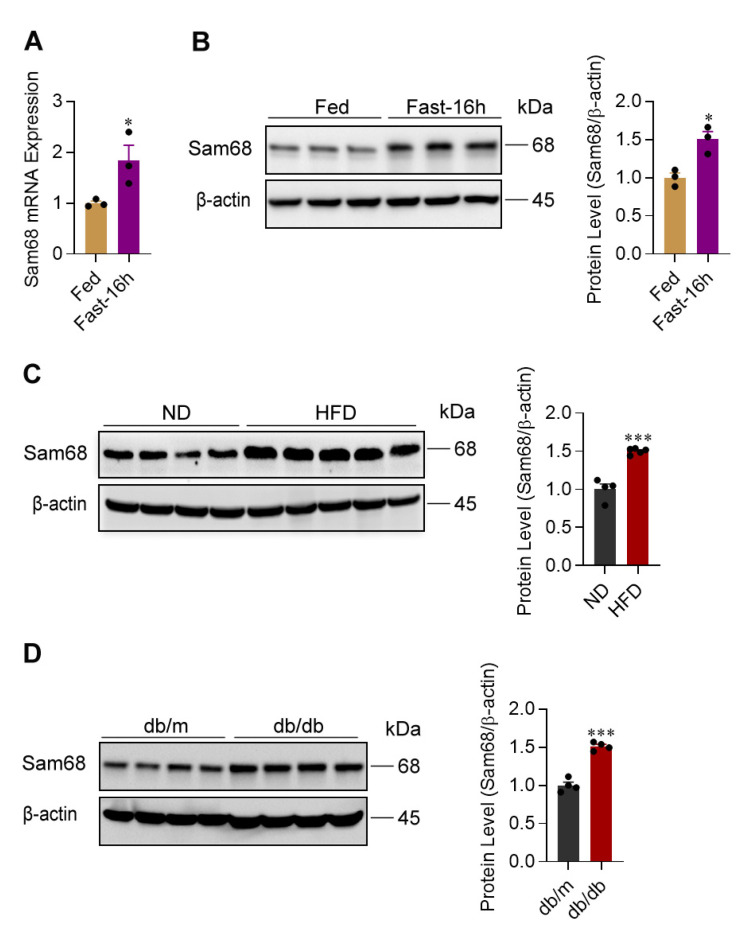
Sam68 is upregulated in the liver of fasting mice, HFD-induced obese and diabetic mice, and db/db diabetic mice. (**A**,**B**) Sam68 mRNA (qRT-PCR) (**A**) and protein ((**B**), left panel, representative Western blot; right panel, quantification) expression were evaluated in the liver of WT mice under feeding condition or after fasting for 16 h. *n* = 3. (**C**,**D**) Sam68 protein expression was evaluated in the livers of (**C**) WT mice on normal diet (ND) or HFD for 12 weeks and of (**D**) db/m (control) and db/db mice at age of 8–12 weeks (left panel, representative Western blot; right panel, quantification). *n* = 4–5. Data are expressed as mean ± s.e.m. * *p* < 0.05, *** *p* < 0.001 (unpaired *t* test).

**Figure 2 ijms-23-11469-f002:**
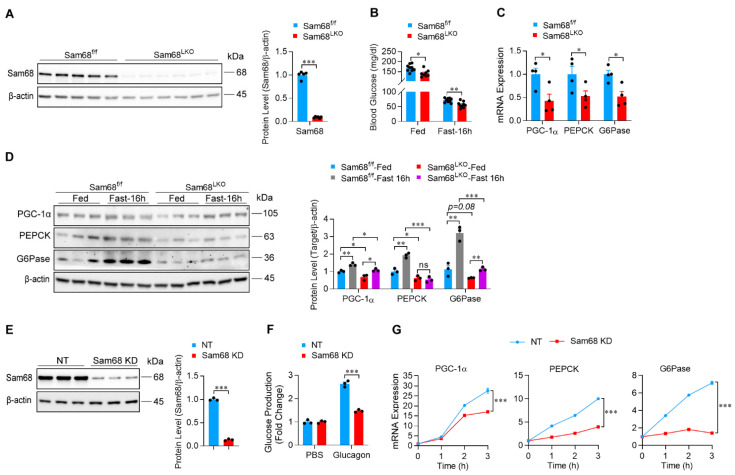
Deletion of hepatic Sam68 attenuates glucagon-induced gluconeogenic gene expression. (**A**) Sam68 protein expression in the liver of Sam68^LKO^ and Sam68^f/f^ mice (left panel, representative Western blot; right panel, quantification). *n* = 5–6. (**B**) Blood glucose levels in the mice under feeding condition or after 16-h fasting. *n* = 8–9. (**C**) mRNA expression (qRT-PCR) of PGC-1α, PEPCK, and G6Pase in the liver of mice after 16-h fasting. *n* = 4. (**D**) Protein expression of PGC-1α, PEPCK, and G6Pase in the liver of mice under feeding condition or after 16 h fasting (left panel, representative Western blot; right panel, quantification). *n* = 3. (**E**–**G**) HepG2 cells were infected with lentiviral vector coding for Sam68-shRNA (KD) or non-targeting shRNA (NT), then the transduced cells were selected in puromycin for 14 days. (**E**) Sam68 protein expression was evaluated (left panel, representative Western blot; right panel, quantification). *n* = 3. (**F**) Glucose production was measured in the culture media of cells that had been treated with glucagon (100 nM) or PBS for 4 h. *n* = 3. (**G**) mRNA expression of gluconeogenic genes was measured in cells after treatment with glucagon (100 nM) for 0–3 h (*n* = 3). Data are expressed as mean ± s.e.m. * *p* < 0.05, ** *p* < 0.01, *** *p* < 0.001, ns, not significant ((**B**,**C**,**F**), and right panels of (**A**,**E**): unpaired *t* test; G and right panel of (**D**): two-way ANOVA).

**Figure 3 ijms-23-11469-f003:**
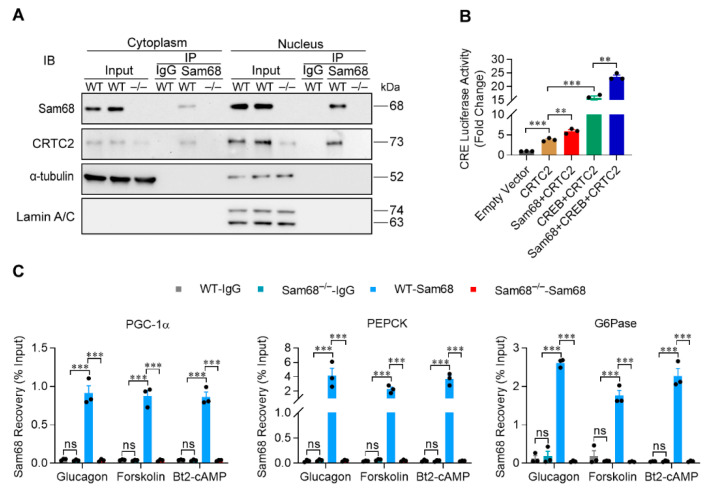
Sam68 interacts with CRTC2 in the nucleus, enhances CREB/CRTC2 transactivity, and co-occupies on the CRE motif in the promoters of gluconeogenic genes. (**A**) WT and Sam68^–/–^ primary hepatocytes were treated with glucagon (100 nM) for 30 min, then their nuclear and cytoplasmic fractions of proteins were isolated; Sam68 protein in each fraction was co-immunoprecipitated, and CRTC2 protein in the precipitates was detected by immunoblotting. (**B**) Plasmids coding for CRE-containing promoter-driven firefly luciferase expression and CMV promoter-driven renilla expression were co-transfected with different combinations of pcDNA3-HA (Empty), pcDNA3-HA-Sam68, pcDNA3-Flag-CRTC2, and/or pCF-Flag-CREB plasmids in HepG2 cells; 48 h later, luciferase activity was measured and normalized to renilla activity. *n* = 3. (**C**) WT and Sam68^–/–^ hepatocytes were treated with glucagon (100 nM), forskolin (10 μM), or Bt2-cAMP (100 μM) for 30 min; then, Sam68 occupancy on the CRE motif of promoters for PGC-1α, PEPCK, and G6Pase were evaluated via ChIP assay. *n* = 3. Data are expressed as mean ± s.e.m. ** *p* < 0.01, *** *p* < 0.001, ns, not significant ((**B**,**C**): two-way ANOVA).

**Figure 4 ijms-23-11469-f004:**
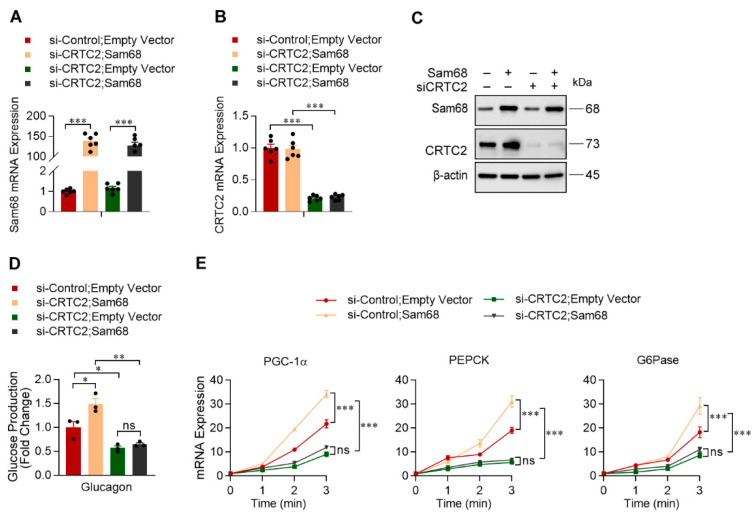
Sam68 promotes gluconeogenesis through CRTC2. pcDNA3-HA-Sam68 or pcDNA3-HA (Empty) plasmid was co-transfected with CRTC2 siRNA or non-targeting siRNA in HepG2 cells; 48 h later, the efficiencies of transfections were evaluated by qRT-PCR for Sam68 (**A**) and CRTC2 (**B**) mRNA expression and by Western blotting for (**C**) Sam68 and CRTC2 protein levels. *n* = 3. The transduced cells were also treated with glucagon (100 nM), then (**D**) glucose produced in the cell-culture media was measured after 4 h of treatment (*n* = 3), and (**E**) mRNA expression of PGC-1α, PEPCK, and G6Pase were evaluated at 0 [basal], 1, 2, and 3 h of treatment (*n* = 3 per time point). Data are expressed as mean ± s.e.m. * *p* < 0.05, ** *p* < 0.01, *** *p* < 0.001, ns, not significant (two-way ANOVA).

**Figure 5 ijms-23-11469-f005:**
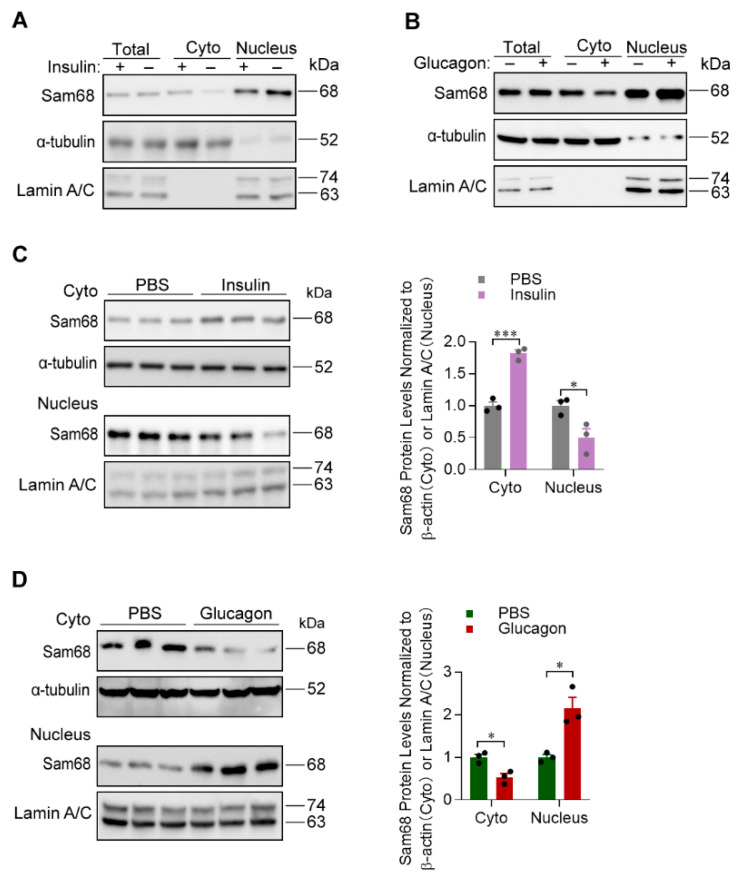
Sam68 shuttling between the cytoplasm and nucleus is regulated by the opposing actions of insulin and glucagon. (**A**,**B**) WT primary hepatocytes were treated with (**A**) insulin (100 nM) or (**B**) glucagon (100 nM) for 30 min; PBS was used as control treatment. Then, whole cellular, nuclear, and cytoplasmic fractions of proteins were isolated, and the abundance of Sam68 protein in each fraction was analyzed by Western blotting. (**C**,**D**) WT mice were intraperitoneally injected with (**C**) insulin (1 U/kg body weight) or (**D**) glucagon (30 µg/kg body weight); PBS was used as control injection. Nuclear and cytoplasmic fractions were isolated from liver tissues 20 min after insulin injection or 10 min after glucagon injection, and Sam68 protein expression in each fraction was evaluated (left panel, representative Western blot; right panel, quantification). *n* = 3. * *p* < 0.05, *** *p* < 0.001, ns, not significant (right panel of (**C**,**D**): unpaired *t* test).

**Figure 6 ijms-23-11469-f006:**
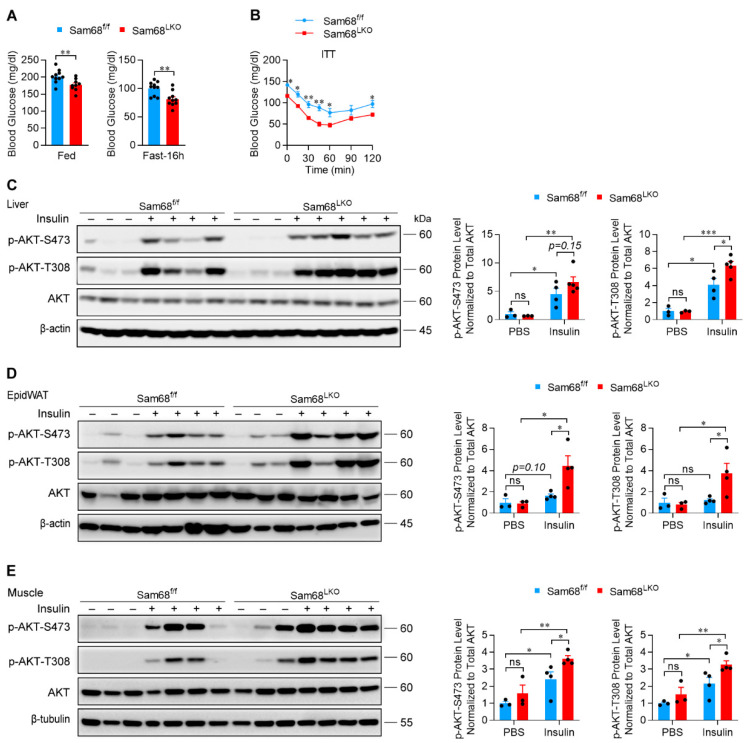
Deletion of hepatic Sam68 improves systemic insulin sensitivity in HFD-fed mice. Four-week-old Sam68^LKO^ and Sam68^f/f^ mice were fed with HFD for twelve weeks and analyzed. (**A**,**B**) Blood glucose levels were measured in mice (**A**) under feeding conditions or after 16 h fasting (*n* = 9–10) and (**B**) in the ITT (insulin tolerance test; *n* = 7–8). (**C**–**E**) Sam68^LKO^ and Sam68^f/f^ mice were intraperitoneally injected with insulin (+) or PBS (–); 20 min later, (**C**) liver (*n* = 3–5); (**D**) epiWAT (*n* = 3–4), and (**E**) skeletal muscle tissues (*n* = 3–4) were harvested, and the protein levels of phosphorylated AKT (at amino acids Ser-473 and Thr-308) and total AKT were evaluated (left panel, representative Western blot; right panel, quantification). Data are expressed as mean ± s.e.m. * *p* < 0.05, ** *p* < 0.01, *** *p* < 0.001, ns, not significant ((**A**): unpaired *t* test; (**B**–**E**): two-way ANOVA).

**Figure 7 ijms-23-11469-f007:**
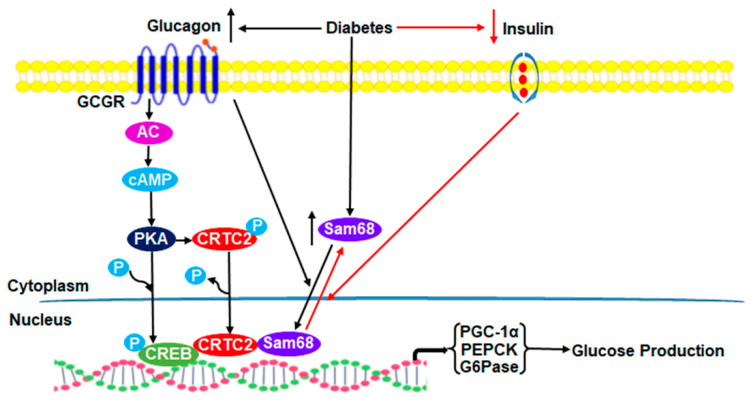
Schematic presentation of molecular mechanisms underlying Sam68 subcellular shuttling and Sam68-mediated hepatic gluconeogenesis. Our results suggest that Sam68 shuttling between the nucleus and cytoplasm is regulated by the opposing actions of glucagon and insulin and plays an important role in the regulation of hepatic gluconeogenesis and glucose production. Specifically, glucagon signaling triggers Sam68 translocation into the nucleus, where Sam68 interacts with CRTC2 to augment CREB/CRTC2 transactivity and gluconeogenic gene expression, promoting glucose production. On the other hand, insulin signaling promotes Sam68 export to the cytoplasm, diminishing CREB/CRTC2 transactivity and gluconeogenesis. In diabetes, hepatic Sam68 expression is elevated, and the skewed metabolic signaling, i.e., increased glucagon signaling and/or decreased insulin signaling, induces Sam68 translocation into the nucleus, where it augments CREB/CRTC2 transactivity, leading to hyperglycemia. GCGR: glucagon receptor; AC: adenylyl cyclase; PKA: protein kinase A; SMEK/PP4C: suppressor of MEK null/protein phosphatase 4 catalytic subunits; PP2B: protein phosphatase 2B.

## Data Availability

All data are presented in the report. Additional details are available on reasonable request.

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
