# Peer review of "Hepatic Sam68 Regulates Systemic Glucose Homeostasis and Insulin Sensitivity"

_ijms, 2022, doi:10.3390/ijms231911469_

Round 1

Reviewer 1 Report

The authors has planned this study very nicely except for few improvement i can ask for.

1. Why the authors have not showed any histomorphological changes in the liver. It will be great if they can show through HE staining. You can refer to the paper and cite "doi: 10.1016/j.molmet.2021.101300". The authors can also try to focus on Immunohistological changes of Hepatic Sam68 in HFD or db/db liver?

2. Do the authors have any future plan to go for bulk RNA seq utilizing these mice liver. It will be a great way to move further or deep molecular mechanism in the sam68 LKO mice.

Author Response

Please see the attachment. Thanks!

Reviewer 2 Report 

The authors investigated the molecular mechanisms underlying in the development of type 2 diabetes from the liver side. The results are clear and contribute to current knowledge significantly. However, some revision must be performed before publishing of the manuscript.

1. The choice of C57Bl, HFD and db/db mice as a relevant model is not clear. Some brief characteristic of this models relevant to performed experiment could add a better understanding of the value of the results. The authors may assume the leptin resistance in db/db mice (PMID 31346871).

2. The description of methods don't contain some important details, such as exact number of animals in each group. The following ARRIVE 2.0 guideline might give greater clarity.

3. In the statistic section, the authors describe that they used parametric tests and present their results as parametric (mean +/- SEM) while they did not mention if they had checked if the distribution of data was normal. 

4. Minor additional corrections into the method section: (1) Clarify the insulin and glucagon (murine or human? supplier?) that they use in the experiment. (2) Double check the references to antibody into Table S2 (e.g. ab74472 is anti-S100A10 but not Sam68 for WB, Sam68 for WB is ab76471).

5. The authors refer quite outdated sources: only 10 of 50 references are published in 2018 and later (5-year period), while 12 references are published in 2013-2017 (10-year period) and 28 of 50 references (more than half) are published in 2012 and earlies (more than 10-year period). Using some modern references, for example PMID: 34914893, 34417460, might be better.

Round 2

Reviewer 1 Report

Thanks for justifying all the queries raised by me.